# Estimation of Global Cropland Gross Primary Production from Satellite Observations by Integrating Water Availability Variable in Light-Use-Efficiency Model

Dandan Du [1,2], Chaolei Zheng [1], Li Jia [1,*], Qiting Chen [1], Min Jiang [1], Guangcheng Hu [1] and Jing Lu [1]

[1] State Key Laboratory of Remote Sensing Science, Aerospace Information Research Institute, Chinese Academy of Sciences, Beijing 100101, China; dudandan19@mails.ucas.ac.cn (D.D.); zhengcl@radi.ac.cn (C.Z.); chenqt@radi.ac.cn (Q.C.); jiangmin@aircas.ac.cn (M.J.); hugc@radi.ac.cn (G.H.); lujing@aircas.ac.cn (J.L.)

[2] University of Chinese Academy of Sciences, Beijing 100049, China

* Correspondence: jiali@aircas.ac.cn

**Abstract:** Satellite-based models have been widely used to estimate gross primary production (GPP) of terrestrial ecosystems. Although they have many advantages for mapping spatiotemporal variations of regional or global GPP, the performance in agroecosystems is relatively poor. In this study, a light-use-efficiency model for cropland GPP estimation, named EF-LUE, driven by remote sensing data, was developed by integrating evaporative fraction (EF) as limiting factor accounting for soil water availability. Model parameters were optimized first using $CO_2$ flux measurements by eddy covariance system from flux tower sites, and the optimized parameters were further spatially extrapolated according to climate zones for global cropland GPP estimation in 2001–2019. The major forcing datasets include the fraction of absorbed photosynthetically active radiation (FAPAR) data from the Copernicus Global Land Service System (CGLS) GEOV2 dataset, EF from the ETMonitor model, and meteorological forcing variables from ERA5 data. The EF-LUE model was first evaluated at flux tower site-level, and the results suggested that the proposed EF-LUE model and the LUE model without using water availability limiting factor, both driven by flux tower meteorology data, explained 82% and 74% of the temporal variations of GPP across crop sites, respectively. The overall KGE increased from 0.73 to 0.83, NSE increased from 0.73 to 0.81, and RMSE decreased from 2.87 to 2.39 g C m$^{-2}$ d$^{-1}$ in the estimated GPP after integrating EF in the LUE model. These improvements may be largely attributed to parameters optimized for different climatic zones and incorporating water availability limiting factor expressed by EF into the light-use-efficiency model. At global scale, the verification by GPP measurements from cropland flux tower sites showed that GPP estimated by the EF-LUE model driven by ERA5 reanalysis meteorological data and EF from ETMonitor had overall the highest $R^2$, KGE, and NSE and the smallest RMSE over the four existing GPP datasets (MOD17 GPP, revised EC-LUE GPP, GOSIF GPP and PML-V2 GPP). The global GPP from the EF-LUE model could capture the significant negative GPP anomalies during drought or heat-wave events, indicating its ability to express the impacts of the water stress on cropland GPP.

**Keywords:** gross primary production; light-use-efficiency model; cropland; water availability; evaporative fraction; ETMonitor

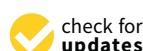



## 1. Introduction

Terrestrial gross primary production (GPP), the amount of carbon assimilation by plants through photosynthesis, is important for the global carbon budget [1]. Accurately quantifying terrestrial GPP is of great importance to evaluating ecosystem carbon dynamics and climate change. Globally, cultivated cropland accounts for about 12% of the land surface, which supplies approximately 15% of carbon fixed and contributes to the large portion of human food [2]. Therefore, the assessment and prediction of agroecosystems

productivity is vital to the agricultural management, crop yield forecast, and global carbon budget assessment [3].

At present, the relatively matured GPP models based on remote sensing satellite data at the canopy scale are mainly divided into two major categories: empirical models and process-based models [4]. The empirical models are relatively simple and easy for regional and global application but have high uncertainty or low accuracy. Most empirical modes are based on the statistical relationship between terrestrial productivity and climate variables or vegetation indexes such as the Normalized Difference Vegetation Index (NDVI), Leaf Area Index (LAI), near infrared radiation from vegetation (NIRv) [5–8], solar induced fluorescence (SIF) [9–11], or Vegetation Optical Depth (VOD) [12–16]. The process-based GPP models are supposed to estimate GPP with higher accuracy because they generally consider detailed ecological processes, but they are difficult to apply at high resolution on global scale due to model complexity and many unknown parameters.

The widely used light-use-efficiency based model (LUE model) [17], mainly driven by remote sensing data, is a semi-empirical model for GPP estimate because it not only considers the response mechanism of photosynthetic efficiency to environmental conditions, but also easily operate on large scales. It can effectively quantify spatiotemporal variation of GPP at regional and global scales if model parameters can be calibrated properly. Many efforts have been made on the calibration and validation of LUE models in regional scale applications [18].

In LUE models, GPP is estimated as the production of absorbed photosynthetically active radiation (APAR) by plant canopy and the efficiency of converting APAR to $CO_2$ fixation through plant photosynthesis [19]. Examples of such models include the Moderate Resolution Imaging Spectroradiometer (MODIS) GPP algorithm (i.e., MOD17) [20], the Vegetation Photosynthesis Model (VPM) [21,22], the light use efficiency model using eddy covariance measurements (EC-LUE) [23], and the Global Production Efficiency Model (GLO-PEM) [24], among others. The differences among these models are mainly the expression of different environmental stress factors used to present the reduction of the potential (or maximal) LUE. The MOD17 algorithm uses the potential LUE (i.e., $\varepsilon_{max}$) of different vegetation types from the biological look-up table and uses two scalars of temperature and vapor pressure deficit (VPD) to account for the reduction in $\varepsilon_{max}$ when cold temperatures and high VPD limit plant function. However, evaluation of the MOD17 GPP by the BigFoot validation approach found that MOD17 GPP seriously underestimated the total GPP in cropland, which was only two-thirds of the GPP estimated by the BigFoot model [25]. Site-level verification studies also showed that MOD17 GPP failed to capture the seasonal and inter-annual changes, with underestimation in peak growing season and overestimation in the early growth and senescence stages [26]. The deviation of MOD17 GPP seasonal variation is mainly due to the inadequate parameterization of moisture constraint [26] and the use of constant value of the potential LUE across crops in different regions.

Soil water availability status is an important factor in GPP modeling. Research reveals that in arid and semi-arid regions, soil moisture is the main controlling factor of GPP, and the substantial impact of soil moisture stress may cause the loss of GPP by 40% in arid/semi-arid and sub-humid regions [27]. In addition, the extent of ecosystems subject to soil moisture stress will further increase in a warming climate in the future [28,29]. However, it is challenging to quantity the influence of soil moisture stress on the estimation of crop LUE over regional or global scales from either modeling or remote sensing observations. Some studies have proposed alternative approaches. For example, the VPM model uses the Land Surface Water Index (LSWI) to estimate the water stress in calculation of LUE, and it performs well in forests [21,22,30]. The Production Efficiency Model Optimized for Crops (PEMOC) uses relative change of Fraction of Absorbed Photosynthetically Active Radiation (FAPAR) across the seasons to estimate seasonal moisture stress, which assumes that the relative change of FAPAR is mainly related to changes in plant water conditions [18]. The EC-LUE model uses evaporative fraction (EF) to calculate moisture stress at the flux tower site-level, and it was found to be reliable for simulating crop GPP [2,23]. EF can

represent moisture conditions of ecosystems because the decrease in energy allocated to the latent heat flux indicates stronger soil water restriction. However, the application of EF for GPP estimation at large spatial scales with moderate spatial resolution (e.g., at kilometer resolution) was hindered by the lack of reliable datasets of sensible and latent heat fluxes at corresponding spatial scales and spatial resolution. Previous studies show that the evaporative fraction based on microwave vegetation indices can help to indicate synoptic-scale water stress on LUE, but limited to relatively coarse resolutions [31]. With the development of global evapotranspiration algorithms and products at higher resolution and improved accuracy [32,33], it is possible to use EF to better express the water availability conditions of ecosystems. Therefore, remote sensing ET or EF is expected to be used to establish the moisture limiting factor in the LUE-based models for global GPP estimation in mid-to-high resolutions.

In addition, the value of model parameters is one of the main issues affecting the accuracy of GPP estimation. Many studies optimize model parameters for specific vegetation types based on ground observations at flux tower sites [34]. However, studies show that some vegetation photosynthetic traits, e.g., the maximum carboxylation rate ($V_{cmax}$), varies in different ecosystems and environmental conditions [35,36]. The fixed value of parameters obtained from the look-up-table specified by plant function types (PFT) is regarded as a major source of uncertainty for GPP estimation because spatial heterogeneity (species composition and plant functional types) is not considered in the LUE models [37–39]. Plant functional type is a classification scheme to reduce the diversity of plant types into the major classes which share similar plant attributes and functions [40]. Some earth system models, such as the global dynamic vegetation model, applies the concept of vegetation functional types [41], but GPP models based on remote sensing rarely consider the plant traits differences across different climate zones. It is important to reduce the uncertainty of model parameters by using ground observations to optimize model parameters for a given plant functional type [42,43], as well as consider the climate conditions.

Therefore, the purpose of this research is to (1) assess the performance of LUE-model for GPP estimation by integrating water availability factor expressed by evaporative fraction (EF) (referred to as the EF-LUE model hereafter) based on site-level ground measurements; (2) optimize the parameters of the EF-LUE model using GPP observations from the eddy covariance flux towers at site-level and extrapolate the optimized parameters spatially according to climate zones for global scale application; and (3) analyze the temporal and spatial characteristics of global cropland GPP estimated by the EF-LUE model and the impact of drought and heat-wave events on cropland GPP.

## 2. Method

### 2.1. Model Description

In this study, *GPP* (g C m$^{-2}$ per time unit) is calculated as the product of *APAR* and actual LUE. The actual LUE is the potential LUE ($\varepsilon_{max}$) constrained by environmental variables including air temperature, VPD, and water availability in a multiplicative manner. *GPP* is then expressed as

$$GPP = APAR \times \varepsilon_{max} \times F_T \times F_{VPD} \times F_W \tag{1}$$

where *APAR* is the absorbed photosynthetically active radiation (MJ m$^{-2}$ per time unit, e.g., hourly, daily, monthly, yearly) and can be estimated as the fraction of photosynthetically active radiation (*PAR*) (MJ m$^{-2}$ per time unit, e.g., hourly, daily, monthly, yearly):

$$APAR = FAPAR \times PAR \tag{2}$$

where *PAR* is generally considered to account for a fraction (e.g., 0.44−0.5) of total solar shortwave irradiance [44,45], 0.48 is taken in this study [46]. *FAPAR* is the fraction of the absorbed photosynthetically active radiation that can be retrieved from satellite observa-

tions of multi-spectral reflectance or derived from NDVI or LAI from satellite observations using empirical relationship.

In Equation (1), $\varepsilon_{max}$ (g C MJ$^{-1}$) is the potential LUE without environmental stresses (also known as maximum LUE), which defines the canopy photosynthetic capacity and varies across vegetation types and environment conditions [47,48]. In this paper, the $\varepsilon_{max}$ values are optimized using ground site data and interpolated according to climate zones for GPP estimate at global scale (see the optimization procedure in Section 2.2).

The environmental constraint factors $F_T$, $F_{VPD}$ and $F_W$ in Equation (1) are dimensionless factors in the range of $0 \leq F_x \leq 1$ with x denoting for T, VPD, or W (water availability), characterizing the constraints of ambient temperature, atmospheric water vapor pressure deficit and soil water availability on the potential LUE. The $F_x$ values closer to "0" indicate stronger constraint by the environmental factor "x", resulting in the reduction of potential LUE and thus lower GPP. On the contrary, $F_x$ values closer to "1" indicate the lower constraint by the corresponding environmental factor "x". $F_x = 1$ means that the constraint is not considered.

The air temperature constraint factor ($F_T$) is usually defined as a ramp function or an asymmetric curve. Using the asymmetric formula, photosynthesis responds more smoothly to air temperature and reverses after the optimal temperature ($T_{opt}$),

$$F_T = \frac{(T - T_{min}) \times (T - T_{max})}{(T - T_{min}) \times (T - T_{max}) - (T - T_{opt})^2} \tag{3}$$

where $T$ is the ambient temperature (°C); $T_{min}$, $T_{max}$, and $T_{opt}$ are the minimum, maximum, and optimum air temperatures (°C) for vegetation photosynthesis, respectively. $F_T$ is set to 0 when ambient temperature $T$ is lower than $T_{min}$ or higher than $T_{max}$, resulting in "0" LUE. In this study, $T_{min}$ and $T_{max}$ are set as 0 °C and 40 °C, respectively. $T_{opt}$ is optimized using GPP that is derived from $CO_2$ flux measurements collected at cropland flux tower sites.

The atmospheric water Vapor Pressure Deficit ($VPD$) is the difference between the saturated water vapor pressure and the actual water vapor pressure at a certain air temperature, indicating the atmospheric water demand. As $VPD$ increases, crops tend to close stomata to reduce the water loss thereafter absorb less $CO_2$. The constraint factor in response to $VPD$ ($F_{VPD}$) is defined as concave functions by

$$F_{VPD} = \frac{VPD_0}{VPD_0 + VPD} \tag{4}$$

where $VPD_0$ is the half-saturation coefficient (kPa) and will be obtained by the optimization procedure described in Section 2.2.

The water availability constraint factor, $F_W$, can be expressed by the evaporative fraction that describes the available energy partitioning to evapotranspiration of vegetation canopy and is controlled by both available energy and availability of water in the soil. The factor $F_W$ is given by

$$F_W = \min\{1, \max(0, EF)\} \tag{5}$$

$$EF = \frac{LE}{LE + H} = \frac{LE}{R_n - G} \tag{6}$$

where $LE$ and $H$ are latent heat flux and sensible heat flux (W m$^{-2}$), respectively. $R_n$ is the net radiation flux (W m$^{-2}$), and $G$ is the soil heat flux (W m$^{-2}$). As our EF-LUE model is applied to daily time scale, the $LE$, $H$, $R_n$ and $G$ are taken as mean daily values.

### 2.2. Optimization of Model Parameters

In this study, the model parameters in the constraint factors, i.e., $\varepsilon_{max}$, $T_{opt}$, and $VPD_0$, were firstly estimated by an optimization procedure using site-level data. The optimization was performed for each site in different climate zones using the Trust Reg ion Reflective algorithm, a non-linear fitting algorithm that is robust to bounded problems, available in

the Scipy.optimize.least_square package (SciPy v1.8.0) in Python 3.6 [49]. It obtains the optimized parameter value by minimizing the sum of squared residuals between the model estimated GPP and the observed GPP. At each site, 2/3 of the available data were used for calibration, and the remaining 1/3 of the data were used for validation. Seed and boundary conditions used for the optimization of the three model parameters are shown in Table 1.

In regional GPP calculation, the optimized parameters were interpolated spatially according to the climate zone classification (see Section 3.1 for data description): for a climate zone with flux tower measurements, the model parameters were optimized using ground measurements from all available sites in the corresponding climate zone; for a climate zone where flux tower measurements are not available, the model parameters were optimized by using the ground measurements from all available cropland sites worldwide (called "Default" values of parameters).

**Table 1.** Seed and boundary conditions of each parameter used for LUE model optimization.

| Parameter | $\varepsilon_{max}$ (g C MJ$^{-1}$) | $T_{opt}$ (°C) | $VPD_0$ (kPa) |
|---|---|---|---|
| Seed | 2.319 | 28.00 | 1.02 |
| Range | [0, 4] | [0, 35] | [0, 3] |

*2.3. Assessment of Optimization and Model Performance*

We used five metrics to assess the model performance, including coefficient of determination ($R^2$), root mean squared errors ($RMSE$), mean predictive error ($BIAS$), Kling-Gupta efficiency ($KGE$), and Nash-Sutcliffe efficiency coefficient ($NSE$). The five formulas are expressed as follows [50,51]:

$$R^2 = \frac{\sum_{i=1}^{N}(O_i - \overline{O})(P_i - \overline{P})}{\sqrt{\sum_{i=1}^{N}(O_i - \overline{O})^2 \sum_{i=1}^{N}(P_i - \overline{P})^2}} \tag{7}$$

$$RMSE = \sqrt{\frac{\sum_{i=1}^{N}(O_i - P_i)^2}{N}} \tag{8}$$

$$BIAS = \frac{\sum_{i=1}^{N}(P_i - O_i)}{N} \tag{9}$$

$$KGE = 1 - \sqrt{(CC - 1)^2 + \left(\frac{P_d}{O_d} - 1\right)^2 + \left(\frac{\overline{P}}{\overline{O}} - 1\right)^2} \tag{10}$$

$$NSE = 1 - \frac{\sum_{i=1}^{N}(O_i - P_i)^2}{\sum_{i=1}^{N}(O_i - \overline{O})^2} \tag{11}$$

where $O_i$ is the observed value, and $P_i$ is the predicted value; $\overline{O}$ and $\overline{P}$ are the mean values of observations and predictions, respectively; $O_d$ and $P_d$ are standard deviation of observations and predictions, respectively. $N$ is the number of total samples to be evaluated. $CC$ is the Pearson coefficient value.

**3. Data**

In this study, several datasets were used, including ground measurements of $CO_2$ flux from the eddy covariance system at flux tower sites of cropland for parameters optimization and validation of the model, satellite data and reanalysis meteorological data as inputs to run the EF-LUE model at global scale, and satellite observation-based GPP products for intercomparison with our model results. The details of these datasets are described in the following sections.

### 3.1. Eddy Covariance Flux Data and Climate Zone Classification Data

Eddy covariance flux data from 19 cropland sites worldwide were downloaded from the FLUXNET [52] (http://www.fluxdata.org, accessed on 25 January 2022), ChinaFlux [53] (http://www.chinaflux.org/, accessed on 25 January 2022), AsiaFlux [54] (http://db.cger.nier.go.jp/asiafluxdb/?page_id=16, accessed on 25 January 2022), and AmeriFlux [55] (https://ameriflux.lbl.gov/, accessed on 25 January 2022). These sites were selected as GPP measurements were available or could be derived from $CO_2$ flux measured by the eddy covariance system over a period of at least three years (Table 2). The spatial locations of these eddy covariance flux tower sites can be seen in Figure 1.

The climate zone classification data used in this study are from the most commonly used climate classification map of Köppen-Geiger with a spatial resolution of 1 km [56]. The Köppen classification method divides the global land surface into five major climate zones based on the relationship between climate and vegetation types, i.e., tropical (A), arid (B), temperate (C), boreal (D), and polar (E). Each one of the five major climate zones is further subdivided into sub-classes (Figure 1) The detailed list of the sub-climate zones Köppen climate classification is given in Table A1 of Appendix A.

**Table 2.** Information of the cropland flux tower sites from which flux measurements by eddy covariance system were collected and used in this study. Climate zone codes were adopted from [56] and given in Table A1 of Appendix A. MAT: mean annual temperature; MAP: mean annual precipitation.

| Site | Latitude | Longitude | MAT (°C) | MAP (mm) | Climate Zone | Period | Crops |
|---|---|---|---|---|---|---|---|
| CH-Oe2 | 47.2863 | 7.7343 | 9.8 | 1155 | Dfb | 2004–2014 | winter wheat/winter barley/rape |
| DaMan | 38.86 | 100.37 | 7.3 | 130.4 | BWk | 2015–2019 | maize |
| DE-Geb | 51.1001 | 10.9143 | 8.5 | 470 | Dfb | 2001–2014 | winter wheat/winter barley/rape/potatoes/summer maize |
| DE-Kli | 50.8931 | 13.5224 | 7.6 | 842 | Dfb | 2004–2014 | winter wheat/winter barley/rape |
| DE-RuS | 50.86591 | 6.44714 | 10 | 700 | Cfb | 2011–2014 | winter wheat/potatoes |
| DE-Seh | 50.87 | 6.45 | 9.9 | 693 | Cfb | 2007–2010 | winter wheat |
| FI-Jok | 60.9 | 23.51 | 4.6 | 627 | Dfc | 2007–2013 | barley |
| FR-Gri | 48.8442 | 1.9519 | 12 | 650 | Cfb | 2004–2014 | winter wheat/winter barley/summer maize |
| IT-CA2 | 42.38 | 12.03 | 14 | 766 | Csa | 2011–2014 | winter wheat |
| MSE | 36.05 | 140.03 | 13.7 | 1200 | Cfa | 2001–2006 | rice |
| US-ARM | 36.6058 | −97.4888 | 14.76 | 846 | Cfa | 2003–2012 | winter wheat/corn/soybean/alfalfa |
| US-Bo1 | 40.01 | −88.29 | 11.02 | 991.29 | Dfa | 2004–2006 | maize/soybean |
| US-Br1 | 41.97 | −93.69 | 8.95 | 842.33 | Dfa | 2009–2011 | corn/soybean |
| US-Br3 | 41.97 | −93.69 | 8.9 | 846.6 | Dfa | 2006–2011 | corn/soybean |
| US-IB1 | 41.86 | −88.22 | 9.18 | 929.23 | Dfa | 2009–2011 | maize/soybean |
| US-Ne1 | 41.1651 | −96.4766 | 10.07 | 790.37 | Dfa | 2001–2013 | maize |
| US-Ne2 | 41.1649 | −96.4701 | 10.08 | 788.89 | Dfa | 2001–2013 | maize/soybean |
| US-Ne3 | 41.1797 | −96.4397 | 10.11 | 783.68 | Dfa | 2001–2013 | maize/soybean |
| YC | 36.83 | 116.57 | 13.1 | 582 | Bsk | 2003–2010 | winter wheat/summer maize |

Eddy covariance systems directly measure $CO_2$ flux (i.e., net ecosystem exchange, NEE) rather than GPP. For the sites from the FLUXNET database, NEE was separated into the two components, GPP and ecosystem respiration, using two different algorithms. The first algorithm was based on the night-time partitioning algorithm [57] where respiration model was parameterized based on the night-time data and then applied to the extrapolation from night to daytime. GPP was then estimated as the difference between respiration and NEE. The second method was the day-time partitioning algorithm [58] where NEE was modelled using the common rectangular hyperbolic light-response curve which was

a function of both GPP and respiration and parameterized based on day-time data. For the sites from database other than FLUXNET, the standardized method provided in the REddyProc online tool [59] was used for processing eddy covariance data. Both night-time and day-time partitioning algorithms were implemented in the tool (https://www.bgcjena.mpg.d-e/bgi/index.php/Services/REddyProcWeb, accessed on 25 January 2022), and the second method was used in this study.

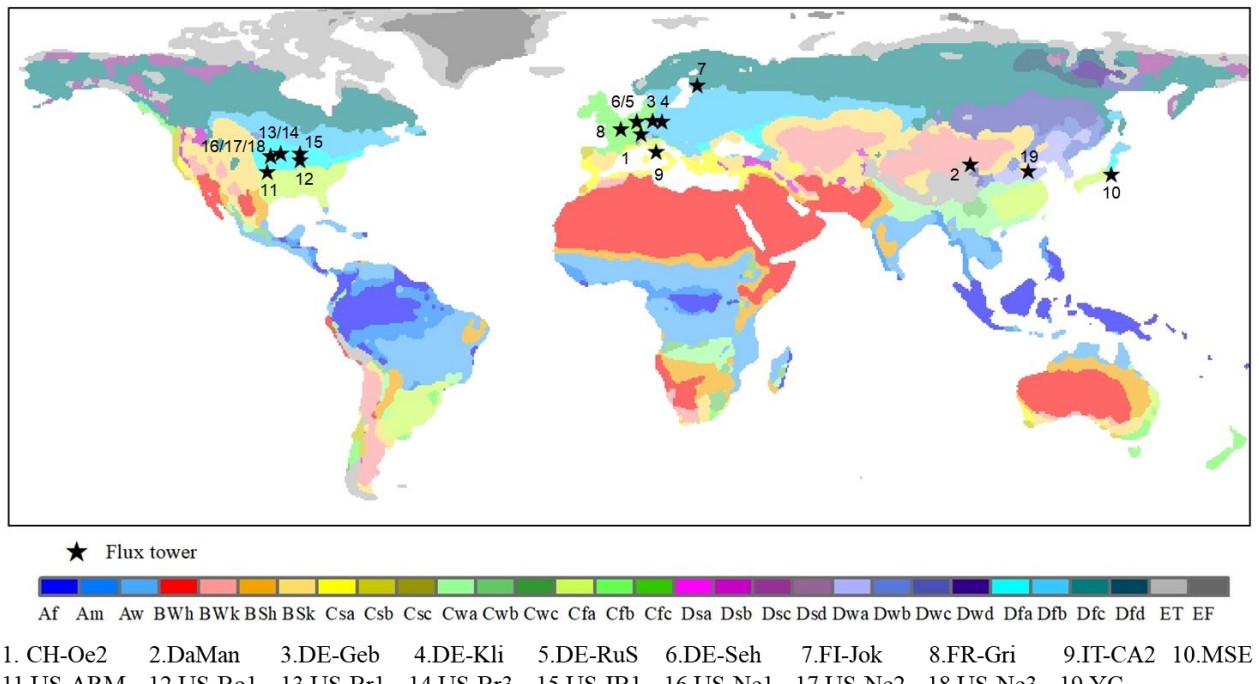

**Figure 1.** The cropland flux tower sites on a map of Köppen climate zones used in the present study. Codes of climate zones in the color legend are adopted from [56] and given in Table A1 of Appendix A. Numbers with star symbol denote the cropland flux tower sites, ground data from which were used in this study.

### 3.2. Meteorological and Remote Sensing Forcing Data

Meteorological and remote sensing input data are showed in Table 3. The meteorological forcing data used in this study, including air temperature, dew point temperature, and downward short-wave radiation flux, were derived from ERA5 (fifth generation ECMWF atmospheric reanalysis) dataset [60] (https://www.ecmwf.int/en/forecasts/datasets/reanalysis-datasets/era5, accessed on 25 January 2022). The ERA5 meteorological forcing data with temporal/spatial resolution of 1-h/0.25° were averaged to daily and downscaled to 1 km resolution using statistical downscaling approaches [61]. Additionally, atmospheric vapor pressure deficit (*VPD*) was calculated from air temperature ($T_a$, °C) and dew point temperature ($T_d$, °C) by:

$$VPD = SVP \times (1 - RH) \tag{12}$$

$$SVP = 0.6112 \times e^{\frac{17.67\,T_a}{T_a + 243.5}} \tag{13}$$

$$RH = e^{\frac{17.625\,T_d}{T_d + 243.04} - \frac{17.625\,T_a}{T_a + 243.04}} \tag{14}$$

where *SVP* represents the saturated vapor pressure (kPa), and *RH* represents the relative humidity (%).

The global-scale remote sensing datasets used as inputs in this study included FAPAR, evapotranspiration (ET) and land cover (LC).

Global annual land cover maps at 300 m spatial resolution from the European Space Agency Climate Change Initiative Land Cover (ESA CCI-LC) project [62] were used and aggregated to 1 km spatial resolution (http://maps.elie.ucl.ac.be/CCI/viewer/download.php, accessed on 23 January 2022). A 1 km pixel was classified as cropland if 50% of the 300 m pixels in the 1 km pixel were cropland.

FAPAR quantifies the fraction of the solar radiation absorbed by vegetation canopy during photosynthesis (Table 3). The 1 km resolution and 10-day global FAPAR product provided by the Copernicus Global Land Service System (CGLS) [63] was used in this study (https://land.copernicus.eu/global/products/fapar, accessed on 23 January 2022).

The global ET data with 1 km spatial resolution and 1-day temporal step were produced (in our parallel study) by the ETMonitor model by using multiple remote sensing data and validated using ground flux measurements at flux tower sites. The detailed description of the ETMonitor model and its applications in regional scales can be found in [32,33,64–67], and a brief introduction to ETMonitor is given in Appendix B. Pixelwise net radiation and surface soil heat flux were estimated in the same way as used in ETMonitor.

**Table 3.** Input datasets used to drive the EF-LUE model for global GPP estimate in this study.

| Variable | Dataset | Resolution | Reference |
|----------|---------|------------|-----------|
| Air temperature (K) | ERA5 | $0.25° \times 0.25°$ 1 h | |
| Dew point temperature (K) | ERA5 | $0.25° \times 0.25°$ 1 h | [60] |
| Surface solar radiation downwards ($Jm^{-2}$) | ERA5 | $0.25° \times 0.25°$ 1 h | |
| Landcover map | ESA CCI | 300 m 1 year | [62] |
| FAPAR | GGLS-GEOV2 | 1 km 10 days | [63] |
| ET, Rn, G | ETMonitor | 1 km 1 day | [32,33,64] |
| Climate classification | Köppen-Geiger | 1 km One map based on data from 1980 to 2016 | [56] |

*3.3. GPP Products by Satellite Remote Sensing Observations*

Five global GPP products derived from satellite remote sensing observations, i.e., MOD17 GPP [20], revised EC-LUE GPP [68], OCO-2-based SIF product (GOSIF) GPP [10], GPP based on near-infrared reflectance of vegetation (NIRv GPP) [7], and GPP from a coupled diagnostic biophysical model − Penman-Monteith-Leuning (PML-v2) model (called PML-V2 GPP) [69], were used for cross comparison with our model results. All these five GPP products provided long-term time series of GPP in global scale and were widely used by researchers. Both the MOD17 GPP and the revised EC-LUE GPP products were produced using LUE-based models, and it would be very interesting to compare these two LUE-based GPP datasets with our EF-LUE model result. Recent studies found that SIF and NIRv performed well in tracking seasonal changes in GPP and were regarded as a proxy for GPP [5,9]. Additionally, the PML-V2 is a water-carbon coupled canopy conductance model that is different in model mechanism from either LUE-based or spectral-index-based methods.

Table 4 gives the summary of the five GPP data products used in this study for cross comparison, and details of each GPP dataset and website for download can be found in the corresponding publications. We used the GPP data from the overlapped years of the five GPP products, i.e., 2002 and 2018, for cross-comparison.

**Table 4.** Satellite observation-based GPP data products used for intercomparison in this study.

| Product Name | Temporal Resolution | Spatial Resolution | Algorithm | Temporal Coverage | Reference |
|---|---|---|---|---|---|
| MOD17 | 8 days | 500 m | LUE | 2000–present | [20] |
| Revised EC-LUE | 8 days | 500 m | LUE | 1982–2018 | [68] |
| GOSIF GPP | 8 days | 5 km | statistical relationship | 2000–2018 | [10] |
| NIRv GPP | Monthly | 5 km | statistical relationship | 1982–2018 | [7] |
| PML-V2 | 8 days | 5 km | canopy conductance | 2002–2018 | [69] |

## 4. Results

### 4.1. Performance of the EF-LUE Model Driven by Ground Measurements at Eddy Covariance Flux Tower Sites

Cropland flux tower sites were grouped according to their locations in the climate zones in Figure 1. The three model parameters, i.e., $\varepsilon_{max}$, $T_{opt}$ and $VPD_0$, were optimized for each climate zone using ground measurements of GPP from all sites in the corresponding climate zone. The parameter optimization procedure was applied to both the EF-LUE model (denoted as 'with EF') and the model without the water availability constraint (denoted as 'without EF'), resulting in two sets of optimized parameters, respectively (Table 5), which would be used in the two LUE models to estimate GPP for comparison. The "All" values (also as 'default') in Table 5 were obtained by the optimization using data from all available cropland flux tower sites listed in Table 2. Obviously, these model parameters varied with different climate zones. In theory, there should be no significant differences in the optimized parameters between using the EF-LUE model and using the model 'without EF' in each climate zone because these three parameters are inherent properties of the crop physiology in response to environmental conditions and should not change with model structures (specifically here for the LUE models 'with EF' and 'without EF'). However, some differences in the parameter values between the model EF-LUE and the model 'without EF' were observed in Table 5, which might be attributed to the uncertainties in the model structure as mentioned by Zheng et al. [70]. The averaged values of $\varepsilon_{max}$ over all climate zones were $2.812 \pm 0.887$ with the EF-LUE model and higher than that used in other models (i.e., 1.044 g C MJ$^{-1}$ in MOD17). The highest $\varepsilon_{max}$ (=3.999) appeared in climate zone 'Bwk' (Table 5) which came from 'DaMan' site with C4 crop (maize) (Table 2). The lowest value of $\varepsilon_{max}$ (=1.506) was in climate zone 'Csa' which was contributed by 'IT-CA2' site with C3 crop (winter wheat). Our values of $\varepsilon_{max}$ parameter are within the range in previous studies found at site-level with $\varepsilon_{max}$ ranging from 2.25 to 4.06 g C MJ$^{-1}$ for C4 crops and from 1.43 to 1.96 g C MJ$^{-1}$ for C3 crops [3,71,72].

**Table 5.** Optimized model parameters ($\varepsilon_{max}$, $T_{opt}$, and $VPD_0$) of the LUE-based model for crop GPP estimate in different climate zones ('with EF' denotes our new EF-LUE model; 'without EF' means water availability constraint was not considered).

| Climate Type | Model without EF | | | Model with EF | | |
|---|---|---|---|---|---|---|
| | $\varepsilon_{max}$ (g C MJ$^{-1}$) | $T_{opt}$ (°C) | $VPD_0$ (kPa) | $\varepsilon_{max}$ (g C MJ$^{-1}$) | $T_{opt}$ (°C) | $VPD_0$ (kPa) |
| CRO/Cfa | 2.999 | 31.037 | 0.590 | 2.725 | 30.655 | 1.262 |
| CRO/Cfb | 2.752 | 18.983 | 0.592 | 2.652 | 17.573 | 1.756 |
| CRO/Csa | 1.506 | 13.306 | 0.478 | 1.509 | 19.475 | 1.651 |
| CRO/Dfa | 2.913 | 35.000 | 2.998 | 3.443 | 34.948 | 2.992 |
| CRO/Dfb | 2.272 | 18.330 | 0.745 | 2.373 | 13.593 | 1.765 |
| CRO/Dfc | 2.249 | 34.964 | 0.759 | 1.959 | 26.274 | 0.646 |
| CRO/BSk | 3.494 | 23.386 | 1.704 | 3.850 | 22.473 | 1.526 |
| CRO/BWk | 3.999 | 32.615 | 1.853 | 3.984 | 29.279 | 1.546 |
| Average | 2.773 ± 0.777 | 25.953 ± 8.506 | 1.215 ± 0.892 | 2.812 ± 0.887 | 24.284 ± 7.260 | 1.643 ± 0.656 |
| All | 2.811 | 34.839 | 2.905 | 2.970 | 29.494 | 2.865 |

The optimized values of the model parameters in each climate zone were applied to the EF-LUE model (Equations (1)–(6)) to estimate GPP driven by the local meteorological variables (EF were calculated using flux measurements at each site) and remote sensing FPAR. For comparison, we also estimated GPP using the model 'without EF' by setting $F_W = 1$ in Equation (1). We compared the model results with the ground measurements at an 8-day time step to examine the model accuracy and capability in capturing the temporal variations of GPP (Figures 2–4). In general, the EF-LUE model and the model without EF driven by tower ground measurements data explained 82 % and 74% of the temporal variations in GPP across the cropland sites, respectively (Figure 2a, only used the validation sub-dataset). The overall Kling-Gupta efficiency (KGE) increased from 0.73 to 0.83, NSE increased from 0.73 to 0.81, and RMSE decreased from 2.87 to 2.39 g C m$^{-2}$ d$^{-1}$ after integrating EF in the LUE model.

Furthermore, the performance of the EF-LUE model and the model without EF for water availability constraint at an 8-day time step were also verified respectively in the summer half year (between the day of vernal equinox (21 March) and the day of autumnal equinox (23 September)) and the winter half year (Figure 2b,c). The accuracy of the EF-LUE model in the summer half year improved significantly over the model without EF, with $R^2$ increased from 0.67 to 0.77 and RMSE decreased from 3.86 g C MJ$^{-1}$ to 3.19 g C MJ$^{-1}$ (Figure 2b). After incorporating EF in the LUE model, the overestimation of low values and the underestimation of high values of the GPP estimated by the LUE model without considering soil water availability were alleviated. In the winter half year, the accuracy of the model was also greatly improved after integrating EF in the LUE model, and the $R^2$ increased from 0.49 to 0.53 (Figure 2c).

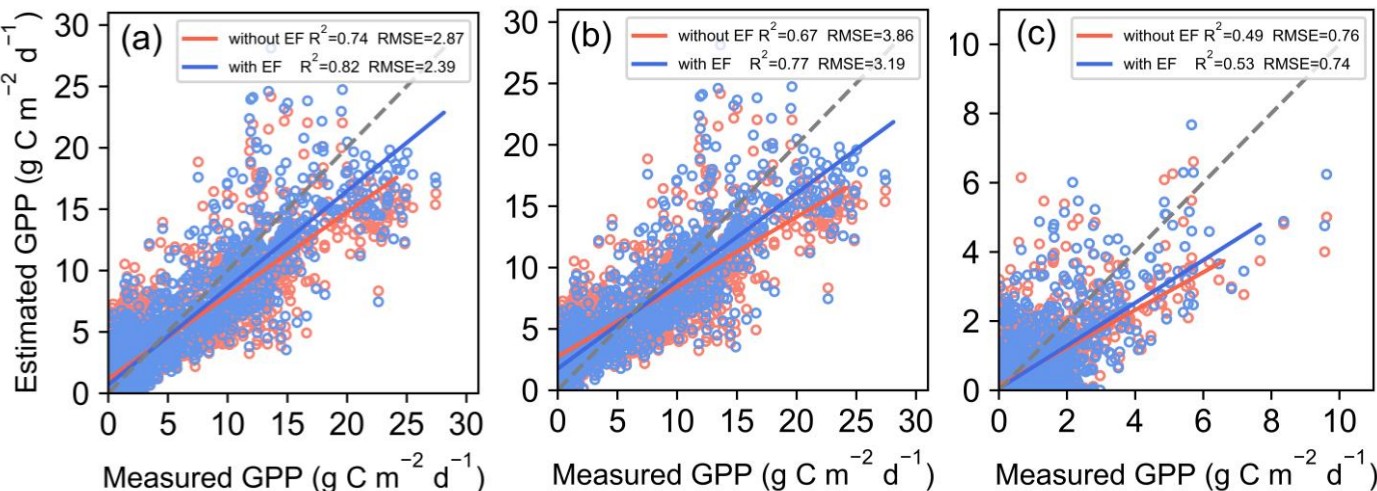

**Figure 2.** Comparisons of GPP between estimation by the models and ground measurements by eddy covariance system at cropland flux tower sites (using validation dataset): (**a**) whole year; (**b**) summer half year (Spring Equinox (21 March) to Autumn Equinox (23 September)); and (**c**) winter half year. EF data were from flux tower measurements. The blue and red points represent GPP estimated by the EF-LUE model and by the model without EF, respectively. The solid lines are the regression lines, and the gray dashed lines are the 1:1 line.

The performance of the EF-LUE model and the model without EF as water availability constraint were also evaluated in different climate zones as shown in Figure 3. The accuracy in the estimated GPP by the EF-LUE model was obviously improved in most climate zones. The results from the EF-LUE model showed the highest $R^2$ in the climate zones of BSk and BWk ($R^2 > 0.9$) (Figure 3a,b). In the Cfa, Cfb, and Dfb climate zones, the improvement of the EF-LUE model was most significant, with an increase of more than 0.1 in $R^2$ over the results of the model without EF (Figure 3c,d,g). In the Csa and Dfc climate zones,

there were too few data samples to give a credible conclusion though both models showed favorable results (Figure 3e,h).

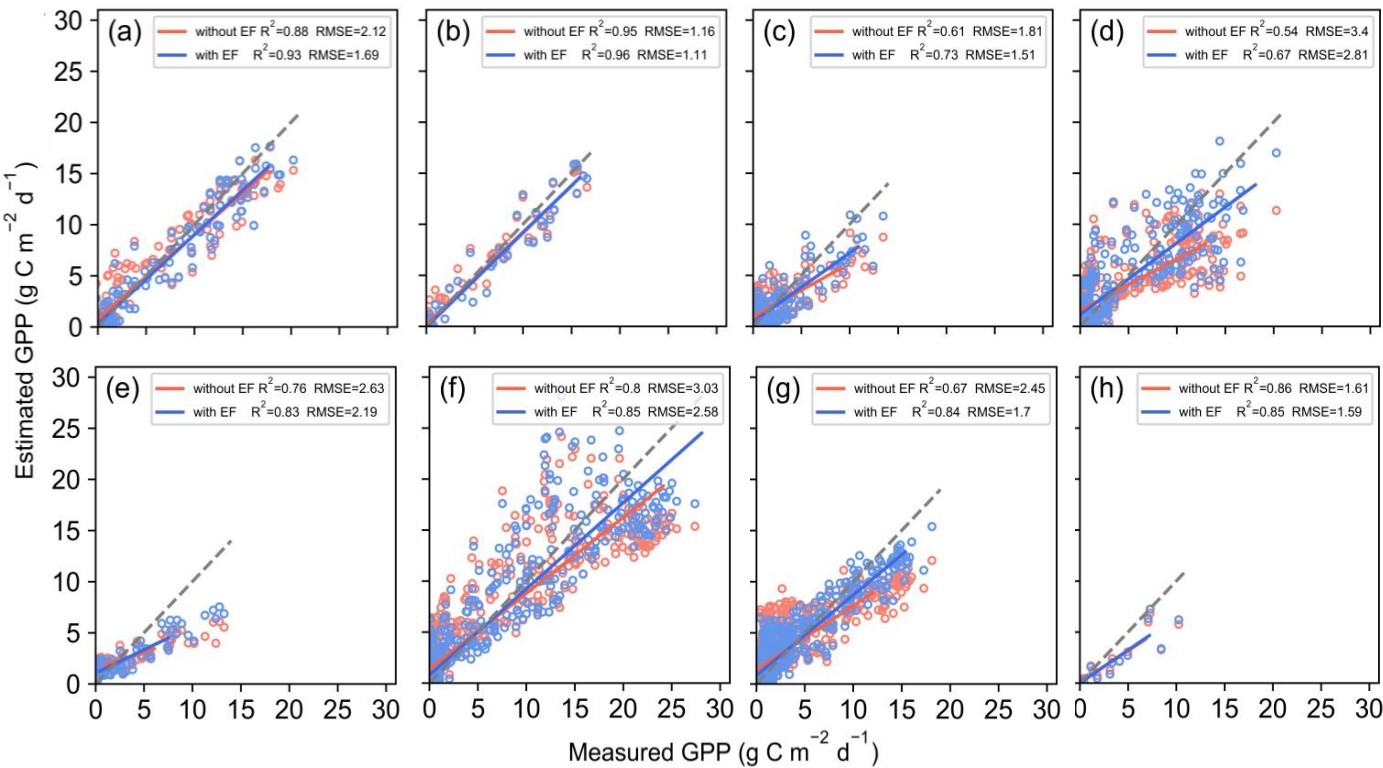

**Figure 3.** Comparisons between GPP estimated by the model and GPP measured by eddy covariance at cropland flux tower sites (using validation dataset) in the following climate zones: (**a**) BSk; (**b**) BWk; (**c**) Cfa; (**d**) Cfb; (**e**) Csa; (**f**) Dfa; (**g**) Dfb; (**h**) Dfc.

Figure 4 gave the results at each flux tower site, the GPP estimated by the EF-LUE model showed higher $R^2$ and lower RMSE in most sites compared with the results by LUE model without EF, especially at CH-Oe2, DE-Geb, DE-Seh, US-Ne2, and US-Ne3. The coefficient of determination ($R^2$) increased from 0.53 to 0.78, and the RMSE decreased from 3.12 to 2.17 at CH-Oe2 sites. The KGE increased from 0.62 to 0.81 at the US-Ne3 site. The NSE increased from 0.73 to 0.88 at the DE-Geb site.

However, although the EF-LUE model generally improved the accuracy in the estimated GPP compared to the method without EF, it performed unsatisfactorily at some sites such as the FR-Gri, US-ARM, and DE-Seh sites ($R^2 < 0.7$) (Figure 4). The results were biased by uncertainty in the satellite-based FAPAR data (figures not shown), which could be improved further by improving the quality of the FAPAR product.

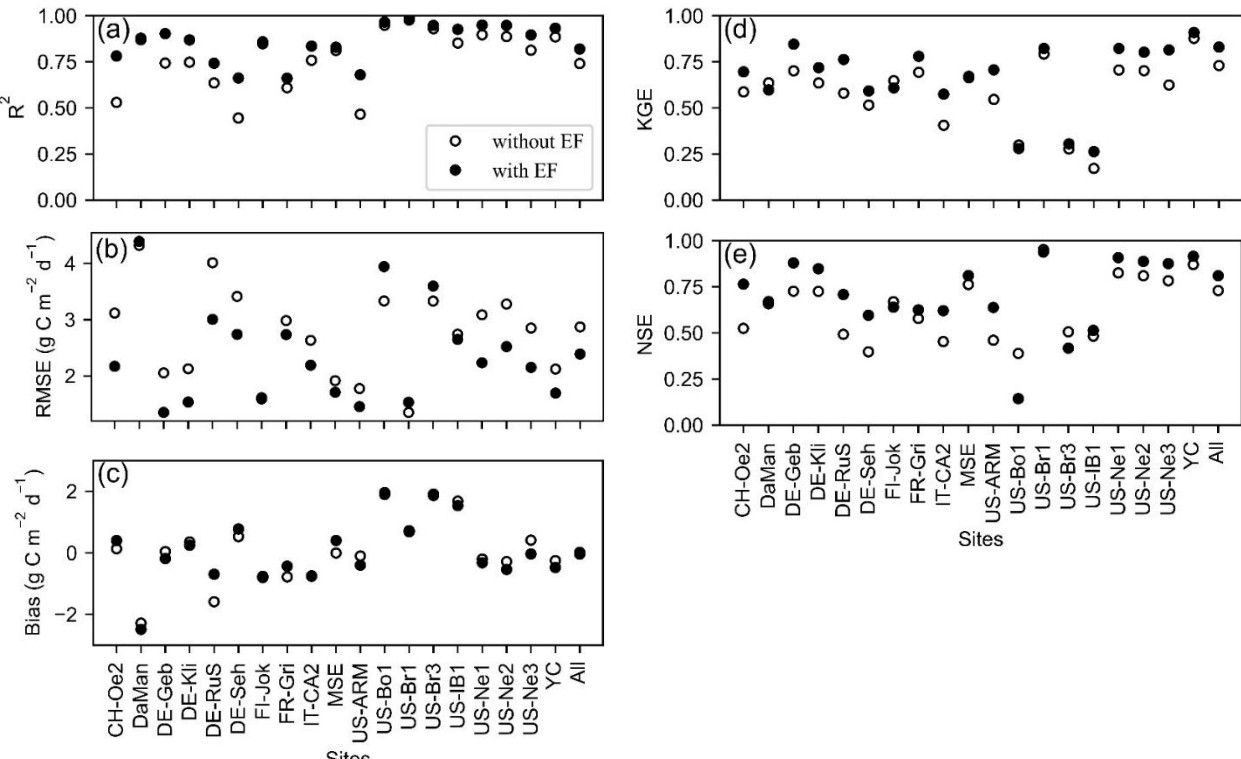

**Figure 4.** Comparisons between GPP estimated by the LUE models (with and without EF) and GPP measured by eddy covariance at flux tower sites (using validation sub-dataset): (**a**) $R^2$; (**b**) RMSE; (**c**) Bias; (**d**) KGE; (**e**) NSE. Solid and open circles represent the results from the EF-LUE model (i.e., with EF) and the LUE model without EF, respectively, driven by meteorological data from flux tower sites.

To further explore the model performance in capturing the temporal variation, we plotted the time series of the estimated GPP together with the ground measurements for the validation sub-dataset at four sites, i.e., DE-Geb, YC, US-ARM, and US-Ne2 (Figure 5). The EF-LUE model could effectively reproduce temporal variations in the GPP observations at these cropland sites. The GPP estimated by the EF-LUE model performed well in tracking the ground-measured GPP at the beginning and the end of the growing season (Figure 5). Moreover, the GPP estimated by the EF-LUE model was closer to the GPP measured by eddy covariance system during the hot summer period at these sites. Soil water availability, expressed by EF, regulated the changes in GPP in the early growing stage and aging stage, as well as during hot summer when soil water stress often occurred. On the contrary, the estimated GPP by the model without EF showed an early start in the beginning of the growing season and a lag in the end of the growing season, in particularly at the site US-Ne2.

However, deviation between the estimated GPP by the EF-LUE model and the measured GPP still existed, which might be attributed to the uncertainties in the forcing data, in particularly the satellite observation-based FAPAR. Another possible source of errors could be the different spatial footprints between the ground flux measurements and the 1 km pixel size for GPP estimate.

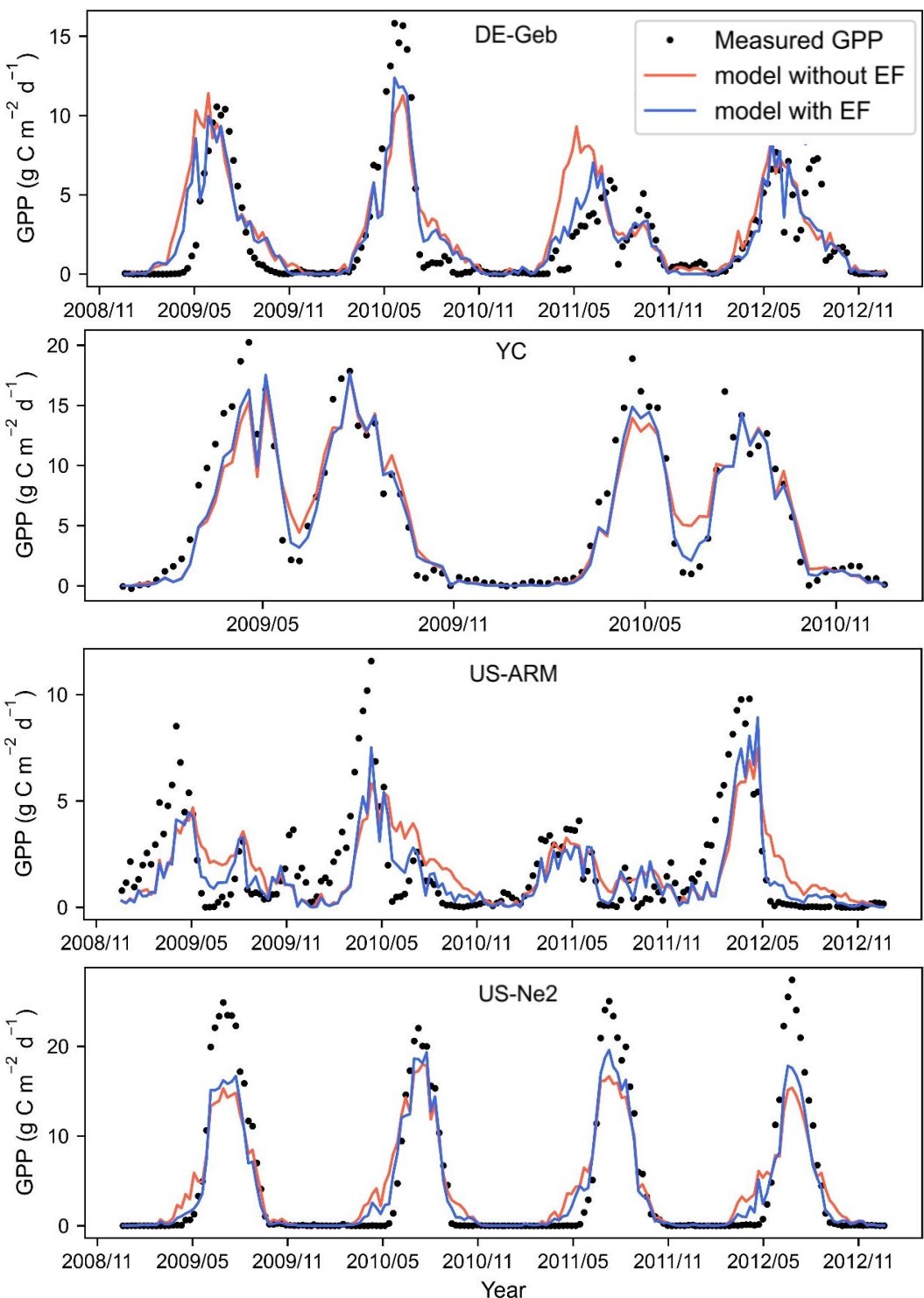

**Figure 5.** Comparisons of 8-day mean GPP between the EF-LUE model and the model without EF against the ground measurements at four cropland flux tower sites. Black solid dots indicate the GPP measurements by eddy covariance system. The blue line and red line represent the GPP estimated by the EF-LUE model and the model without EF, respectively.

## 4.2. Comparison with Other GPP Products at Eddy Covariance Flux Sites

Furthermore, we estimated global GPP at daily and 1 km resolution using the EF-LUE model developed in this study by taking the global gridded data listed in Table 3 as forcing, and compared with other four exiting GPP products (listed in Table 4) for the period of 2002–2018. We first compared the estimated GPP by the EF-LUE model with other

two LUE-based GPP products, i.e., MOD17 GPP [20] and the revised EC-LUE GPP [68], the spectral index based GPP (i.e., GOSIF GPP) [10] and the conductance model based GPP (i.e., the PML-V2 GPP) [69] at an 8-day step for each crop site. Figure 6 showed the boxplot to compare the performance of different GPP products. Generally, the EF-LUE GPP showed the highest $R^2$, KGE, and NSE and the lowest RMSE. These improvements may be attributed to the optimized potential LUE values (i.e., $\varepsilon_{max}$) in different climate zones and the integration of water availability constraint factor (i.e., EF) in the EF-LUE model. The MODIS algorithm used the VPD-based slope function as surrogate for soil moisture status, which partly explained that MOD17 GPP had a higher deviation during severe summer drought caused by the low soil water content [73]. VPD is an effective indicator of atmospheric water demand and might not be sufficient to represent the condition of soil water availability and the effect of soil water stress on the vegetation photosynthetic efficiency. The revised EC-LUE model took into account the effects of diffuse radiation and atmospheric carbon dioxide concentration, but the improvement was limited by the exclusion of EF to reflect the impact of soil water stress on cropland GPP estimation in global scale application.

The performances of GOSIF GPP and PML-V2 GPP were also showed in Figure 6 for a broader comparison, and these two GPP products were claimed to have high accuracy. The EF-LUE GPP showed comparable or better behavior when compared with these two GPP products, e.g., EF-LUE GPP had the overall highest $R^2$ and KGE and lower RMSE, indicating the good ability of the EF-LUE model in global GPP estimation. These results also suggest that the LUE-based model (especially the EF-LUE model in this study) is still very promising for global cropland GPP estimation with satisfactory accuracy if the model is properly structured and model parameters are carefully calibrated, and it retains the advantage of easy application when comparing with the complex process-based models.

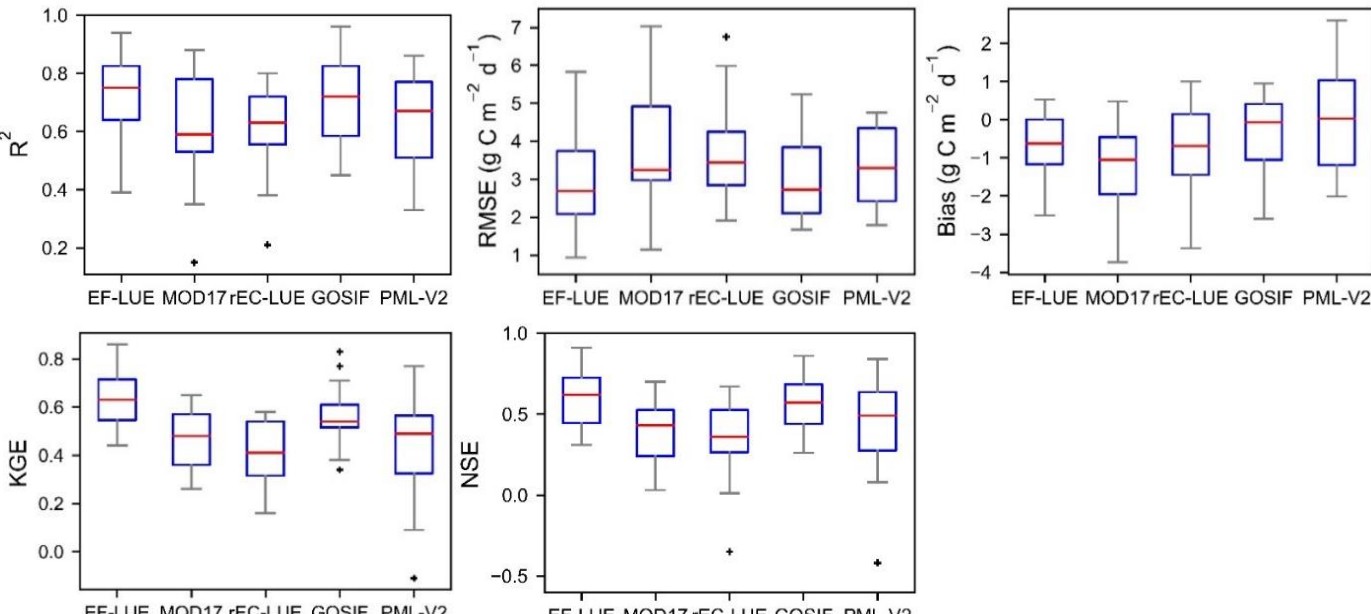

**Figure 6.** Boxplot for comparisons of 8-d mean GPP between the EF-LUE GPP, MOD17 GPP, revised EC-LUE (rEC-LUE in the figures) GPP, GOSIF GPP, and PML-V2 GPP with the GPP measurements at cropland flux tower sites in 2002–2018.

### 4.3. Spatiotemporal Patterns of Global Cropland GPP

We compared the mean annual GPP estimated by the EF-LUE, MOD17 product, the revised EC-LUE GPP product, GOSIF GPP product, PML-V2 GPP product, and NIRv GPP product in the common available years of the six data sets (2002–2018). Spatial patterns of global cropland GPP estimated by the six data sets were similar (Figure 7). The

highest cropland GPP (>1500 g C m$^{-2}$ yr$^{-1}$) was mainly distributed in tropical regions (e.g., Southeast Asia, and Brazil), due to sufficient hydrothermal conditions. The lowest crop GPP regions occurred in Russia and South of the Sahara Desert (<500 g C m$^{-2}$ yr$^{-1}$) where climate and edaphic conditions were rigid, and the level of agricultural development was relatively low.

The global mean annual GPP (2002–2018) for cropland simulated by our EF-LUE model is 1054.8 g C m$^{-2}$ yr$^{-1}$, very similar to the values of 1094.6 g C m$^{-2}$ yr$^{-1}$ from previous studies which is the median value of ground measurements and various diagnostic models (1998–2005) [74]. In general, our result of global mean annual GPP averaged over 2002–2018 was lower than that of the GOSIF and the PML-V2 GPP, and higher than that of the MOD17, the revised EC-LUE, and the NIRv GPP products.

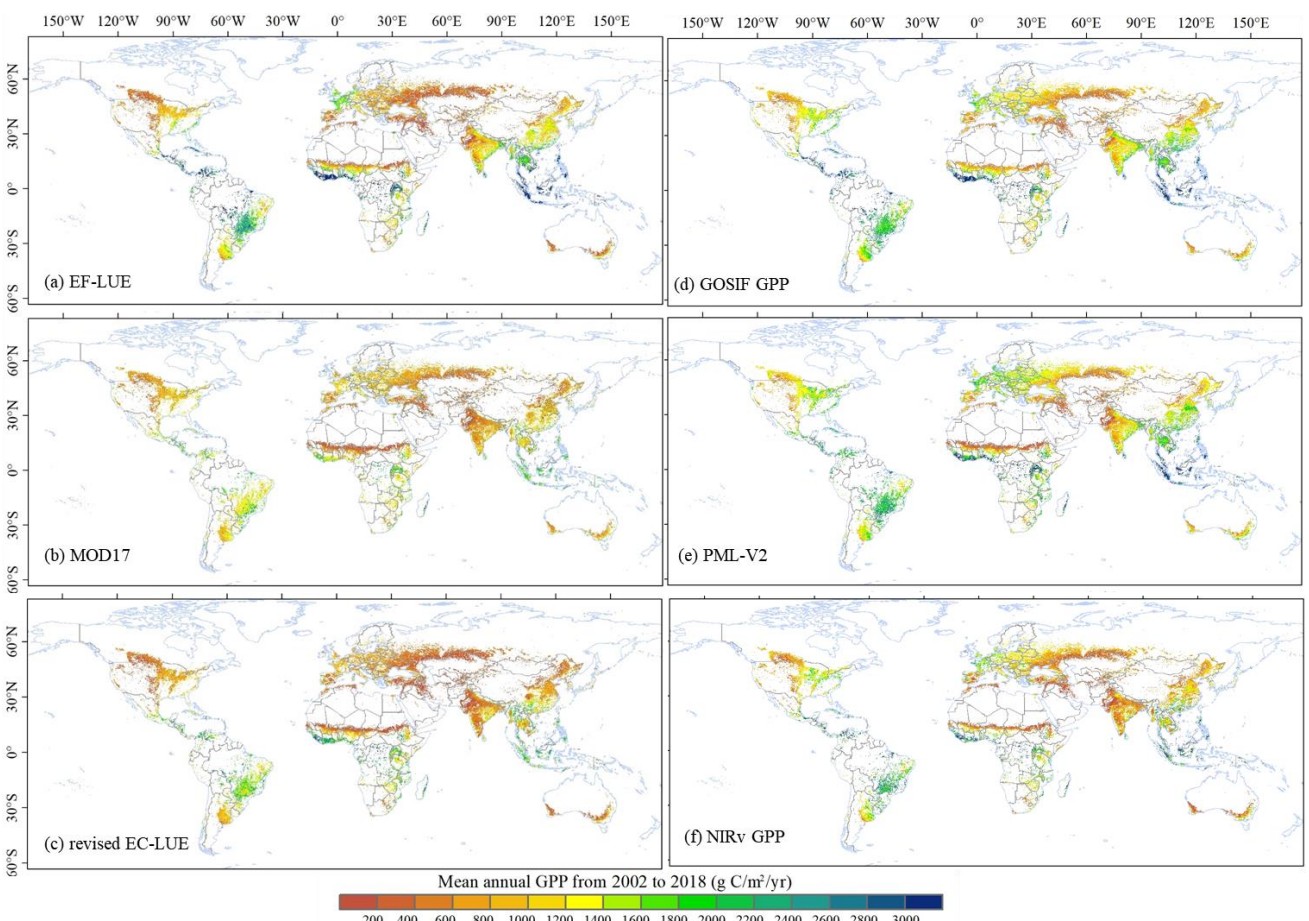

**Figure 7.** Spatial pattern of global mean annual GPP of cropland (average over 2002–2018) estimated by (**a**) EF-LUE, (**b**) MOD17, (**c**) revised EC-LUE, (**d**) GOSIF GPP, (**e**) PML-V2 GPP, and (**f**) NIRv GPP.

Figure 8 showed the difference in mean annual GPP (2002-2018) between the EF-LUE model and the five GPP datasets listed in Table 4. The MOD17 GPP showed generally low values (this was consistent with the site-level assessment where MOD17 gave the largest negative bias as shown in Figure 6), especially in Western Europe, East Asia, Southeast Asia, and some high-productivity regions in Africa and South America, whereas it was higher than the EF-LUE GPP in Central Europe, and Western Russia. This might be attributed to the constant $\varepsilon_{max}$ value for all cropland used in the MOD17 method. Another reason for the large deviation in arid and semi-arid regions might be the insufficient expression of soil moisture constraints in the LUE model of the MOD17 GPP method. The EF-LUE GPP was significantly higher than the other two LUE GPP products and close to GOSIF GPP, NIRv GPP and PML-V2 GPP in Western Europe. However, in Eastern Europe and Russia, our model results seemed lower than GOSIF GPP, NIRv GPP, and PML-V2 GPP,

the PML-V2 GPP showed the highest values in this region. Both the MOD17 GPP and the revised EC-LUE GPP used the VPD function to constrain actual LUE without explicit consideration of soil moisture condition. The values from the revised EC-LUE GPP were generally higher than that of MOD17 GPP in the local high-productivity regions of southern China, Brazil, and Africa, which may be due to the consideration of diffuse radiation in estimation of LUE in the revised EC-LUE method. PML-V2 also used VPD to regulate canopy conductance and GPP. Although SIF is sensitive to drought, uncertainty might be introduced due to the fact that the long-term GOSIF GPP product was generated based on coarse-resolution SIF observations and re-produced at high spatial resolution using the relationship between SIF and other explanatory variables such as EVI, temperature, PAR, and VPD, rather than directly using SIF observations at higher spatial resolution. The NIRv GPP, based only on reflectance from red and near-infrared wavelength, might have missed subtle changes in photosynthetic activity caused by physiological processes associated with pigmentation [6,9].

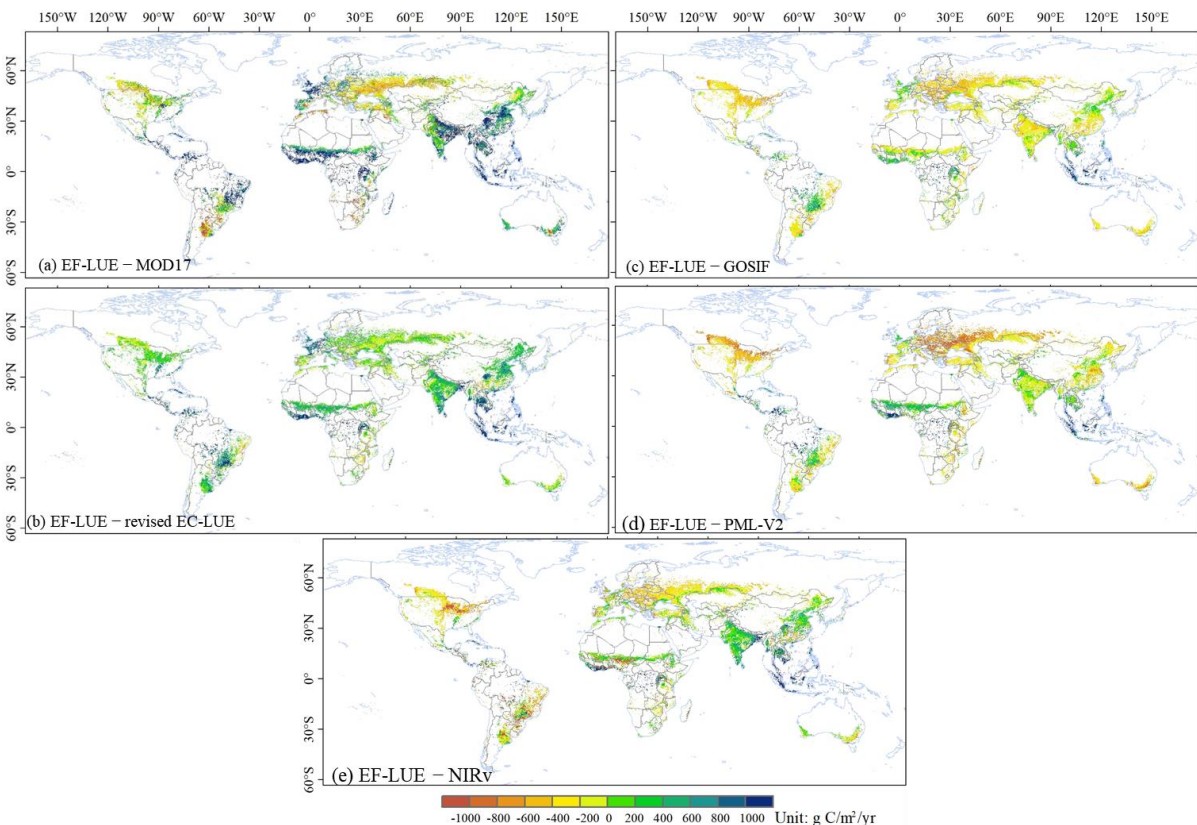

**Figure 8.** Mean annual GPP (in 2002–2018) difference between the EF-LUE model and (**a**) MOD17; (**b**) revised EC-LUE; (**c**) GOSIF GPP; (**d**) PML-V2; (**e**) NIRv GPP.

### 4.4. Comparison of GPP Products in Rainfed and Irrigated Croplands

In order to further evaluate the EF-LUE GPP, we cross-compared our estimated GPP for rainfed and irrigated croplands with the existing GPP products listed in Table 4 (Figure 9). As seen in Figure 9, the global average of GPP estimated by different models from 2002 to 2018 in the rainfed crop is generally higher than that in the irrigated crop for all the models/products. This is consistent with previous study that showed higher rainfed croplands GPP values than that of irrigated croplands [75]. Our GPP results in rainfed cropland are very close to the mean values of the six GPP products, while the GPP in irrigated cropland from our model is slightly higher than the average of the multiple GPP products of irrigated cropland.

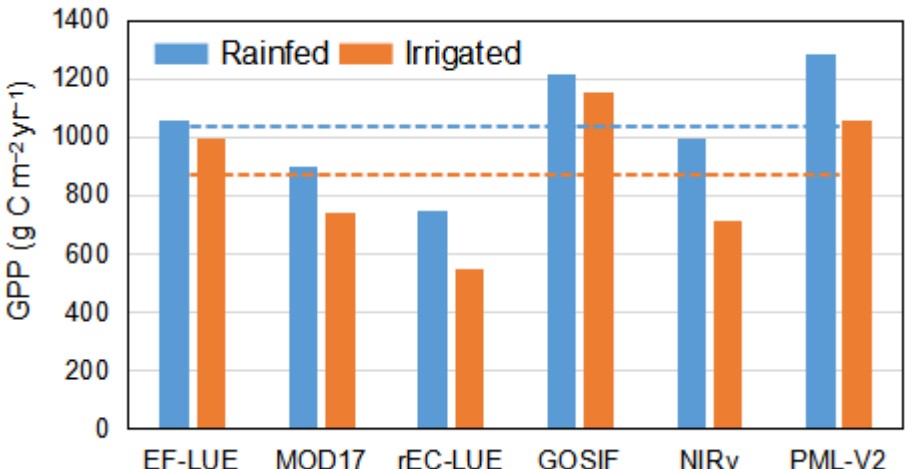

**Figure 9.** Comparison of annual mean GPP simulated by the EF-LUE model with other five different GPP products in rainfed and irrigated croplands from 2002 to 2018. The blue and orange dashed lines are the average GPP values of all 6 GPP products over rainfed cropland and irrigated croplands, respectively.

We compared the average values of GPP estimated by different models for rainfed cropland and irrigated cropland in 5 climate zones (Figure 10). In tropical climate zone (A in Figure 10), the EF-LUE GPP (both rainfed and irrigated cropland) was closer to the GPP values from the GOSIF and the PML-V2 GPPs. Except MOD17, rainfed cropland GPP values by the other five GPP products were higher than the irrigated cropland GPP and significantly higher than the MOD17 rainfed cropland GPP in tropical climate zone. In arid climate zone (B in Figure 10), the EF-LUE GPP of rainfed cropland was closer to the rainfed cropland GPP by GOSIF, MOD17, and PLM-V2. In temperate climate zone (C in Figure 10), the EF-LUE GPP of rainfed cropland was close to that by MOD17, GOSIF, NIRv, and PML-V2, while the irrigated cropland GPP by the EF-LUE model was close to that by GOSIF and PLM-V2. In boreal climate zone (D in Figure 10), the irrigated cropland GPP by the EF-LUE model was close to that by GOSIF, NIRv, and PLM-V2, but the EF-LUE model showed lower GPP in rainfed cropland than MOD17, GOSIF, NIRv, and PLM-V2. In the polar climate zone (E in Figure 10), the GPP values from our EF-LUE model and from the other two LUE-based models were significantly lower than the values from GOSIF and PLM-V2, particularly for irrigated cropland.

In summary, in most climate zones (except the polar zone), our GPP estimation for both rainfed and irrigated croplands were close to that of GOSIF. The previous study found strong positive correlations between SIF and root-zone soil moisture [76]. Proper consideration of the restriction of soil moisture availability as done in our method might help improve the estimation accuracy in cropland GPP. The PML-V2 GPP value in irrigated cropland in dry climate areas was lower than our results and GOSIF GPP, which might be partially due to that the PML-V2 model did not explicitly consider soil moisture constraint in GPP calculation. The PML-V2 GPP was generally higher, but the PML-V2 GPP value in irrigated cropland in dry climate areas was lower. In the arid and boreal climate zones, the revised EC-LUE was significantly lower than other GPP products. A previous study of product comparison also found that VPM and GOSIF showed similar results in cropland GPP estimations whereas MOD17 and the revised EC-LUE GPP had relatively lower values [77].

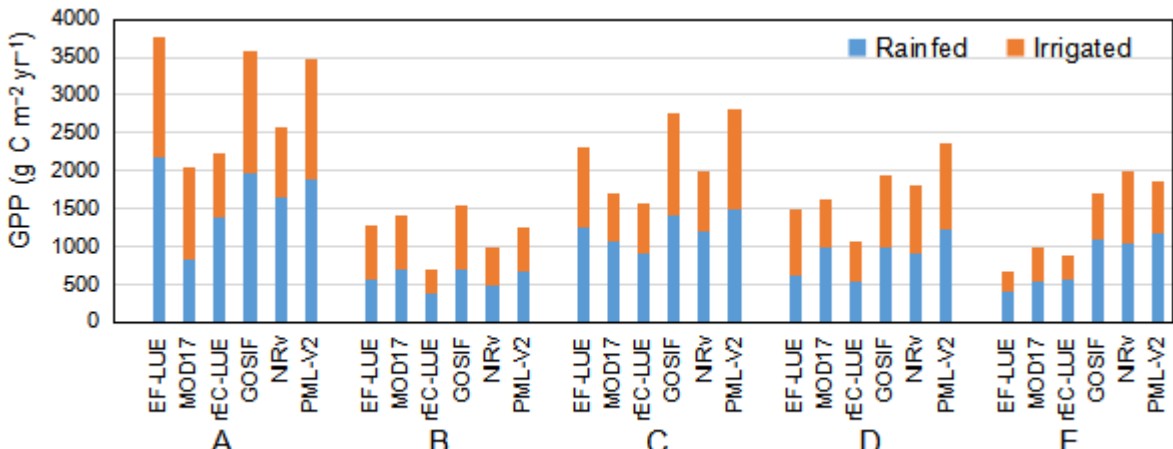

**Figure 10.** Comparison of annual mean GPP (2002–2018) from the EF-LUE model with other 5 GPP products in rainfed and irrigated croplands in tropical (**A**), arid (**B**), temperate (**C**), boreal (**D**), and polar (**E**) climate zones.

*4.5. Assessment of GPP in Response to Extreme Events*

To further evaluate the ability of the EF-LUE GPP to capture the impact of drought and heat-wave events, we analyzed the response of the estimated GPP by the EF-LUE model during different drought/heat-wave events. Four severe heat-wave and drought events that occurred in recent years, including the 2010 Russian heat-wave, 2009 India heat-wave, 2013 China drought, and 2012 US Corn Belt drought, were selected for analysis. Figure 11 showed the percentage of mean GPP anomalies in the period during the heat-wave and drought events with respect to multi-year monthly mean GPP over the same months of extreme events in the past two decades. The heat-wave that occurred in western Russia in the summer of 2010 resulted in the warmest July since 1880 in many regions of Russia [78], which increased sensible heat flux by about 10–15% and reduced latent heat flux by about 15–20% reported in the literature [79]. The impact of heat-wave on the GPP estimated by the EF-LUE model was pronounced with more than 30% reduction in GPP in most croplands as seen in Figure 11a. In 2009, a heat-wave occurred in Orissa, West Bengal, Bihar, Uttar Pradesh, Jharkhand, and Andhra Pradesh provinces in India. Our results showed a significant reduction in GPP during the Indian 2009 heat-wave period (Figure 11b). In 2013, different severe level of droughts occurred in Southwest China, the North China Plain, and south of the Yangtze River, resulting in crop yields reduction to the largest negative anomaly level since 1960 in these regions [80]. GPP estimated by the EF-LUE showed a significant decrease (10~20%) in part of the North China Plain (Figure 11c). In 2012, a severe drought occurred and spread over almost two thirds of the continental US, particularly in the Corn Belt [81], which resulted in 25% reduction in maize yields [82]. Our result showed a significant reduction in GPP (~30%) (Figure 11d).

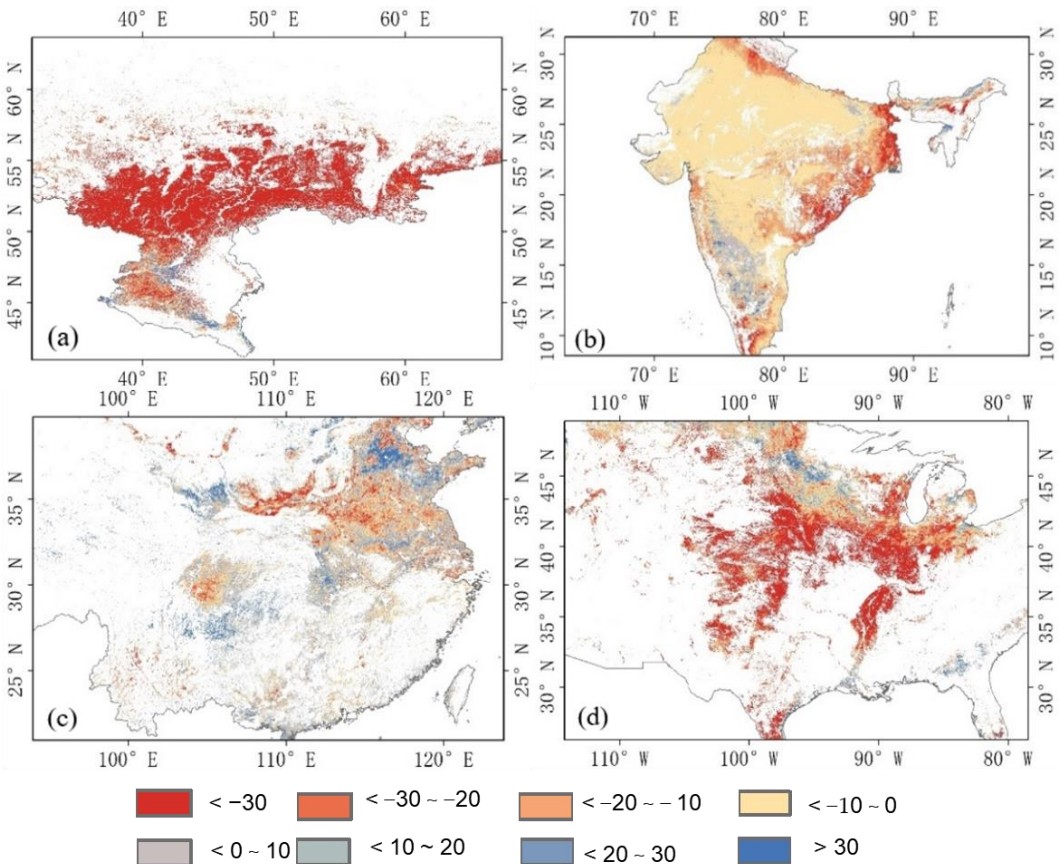

**Figure 11.** The percentage of anomalies of monthly mean GPP relative to monthly GPP of multi-year mean values for four major heat-wave and drought events: (**a**) 2010 (June–August) Russia heat-wave; (**b**) 2009 (April–June) India heat-wave; (**c**) 2013 (January–June) China drought; (**d**) 2012 (June–December) US Corn Belt drought.

## 5. Discussion

The LUE-based models are widely used in remote sensing-based methods for GPP estimation. Actual LUE highly varies and is determined by complex processes of crop physiology and environmental conditions, thus leading to difficulty in GPP estimation with high accuracy at the global scale [83]. Actual LUE is determined by potential LUE ($\varepsilon_{max}$) and environmental conditions. Previous studies suggest that the potential LUE ($\varepsilon_{max}$) is related to different vegetation types and is relatively more stable than actual LUE. However, a study based on flux tower measurements showed the large spatial variability in $\varepsilon_{max}$ within individual biome types [38]. Unlike the MOD17 GPP method which assumed a constant $\varepsilon_{max}$ for all crops, in the EF-LUE model proposed in this paper, the model parameters were optimized using available cropland flux tower data and took into account the parameter differences in different climate zones when applied at global scale. This overcomes the serious underestimation in MOD17 GPP. Based on a literature survey, the range of $\varepsilon_{max}$ was between 0–4 g C MJ$^{-1}$ [83,84]. Our optimized value clearly showed the large range of $\varepsilon_{max}$ values for different crops in different climate zones, with the largest $\varepsilon_{max}$ value (3.9 g C MJ$^{-1}$) found at maize (C4 crop) site. Maize is a C4 crop with a greater LUE than C3 crops, while the value of $\varepsilon_{max}$ for maize derived from ground observations in previous studies was 3.84 ± 0.08 g C MJ$^{-1}$ [85], close to our estimation.

The great variation of cropland LUE is associated with crop types and environmental and water supply conditions, and therefore is dependent on the actual crop cultivation. Compared with natural ecosystems, crop types may change rapidly due to management practices by farmers (such as crop rotation plans), which means the values of $\varepsilon_{max}$ will change with the different crop rotation cycles. There are some regions adopting rotation

or intercropping system, e.g., planting maize and soybeans in different years or in the same year by intercropping. Due to different photosynthesis pathways, the $\varepsilon_{max}$ of maize is much higher than that of soybean. Optimizing the model parameters using the flux tower data without distinguishing the two crops may lead to underestimation of maize GPP and overestimation of soybean GPP. For the cropland GPP estimation based on the LUE type models using remote sensing data at regional and global scales, further attention should be paid to the impact of different crop types (at least distinguishing C3 and C4 crops) and crops phenological periods on the optimization of model parameters. The model parameters can be estimated more precisely if detailed crop types are known.

In the EF-LUE model, the water availability constraint factor, represented by EF, modulates the influence of soil water supply on crop productivity and can avoid overestimation of GPP when crop experiencing water stress. As in previous studies carried out at site-level, EF can be used to quantify soil water availability in LUE models and can improve the accuracy of GPP estimation [23,86,87], and many studies have tried to revise the GPP model based on remote sensing EF as an indicator of water stress at regional scales [88,89]. Soil heat flux was neglected in the calculation of available energy in the study by [88], instead Rn was used in the denominator of the equation for EF calculation in their LUE model for GPP estimation in regional scale application. However, neglecting soil heat flux (G) in the EF calculation may bring large errors in particular over regions where daily soil heat flux is large. For global application in our study, daily EF was calculated by daily ET and daily values of Rn−G from the ETMonitor model, in which daily G was estimated by a machine learning algorithm established based on measurements of G and several explanatory variables, and the results of G showed good agreement with the ground measurements at flux tower sites. In addition, ETMonitor could consider the impact of soil water content on ET, which made it able to capture the spatiotemporal variation of EF well [64,90]. The ETMonitor model was applied to regional scales successfully and proven to have high accuracy as shown in several studies [65–67]. For example, a study on evaluating satellite-based ET products showed that ETMonitor had the highest R value of 0.83 and the lowest RMSD of 0.92 mm/d, while the other 6 ET products had correlation coefficients of 0.58 to 0.78 and RMSD of 1.13 to 1.41 mm/d [91]. Although the accurate ET from ETMonitor partly ensured the overall good performance of the EF-LUE model for cropland GPP estimation in this study, future work is still needed on systematic analysis of the remaining uncertainties brought by the EF.

Carbon and water fluxes are inherently tightly coupled through plant stomatal activity. Carbon gain through photosynthesis and water loss through transpiration are the two dominant processes in the global water and carbon cycles and are simultaneously controlled by plant stomatal behavior in response to environmental conditions [92]. The two processes have similar characteristics to respond to changes in many environmental conditions, such as radiation, temperature, and VPD [93]. Soil moisture affects evapotranspiration and photosynthesis by regulating leave water potential and thereby affecting stomatal conductance that determines carbon and water exchanges between plants and the surrounding atmosphere. Some studies based on the conductance method have used some form of carbon–water coupling to estimate GPP and evapotranspiration [69]. As demonstrated by our study, integrating EF in the LUE type of model for GPP estimation is an effective method to provide satisfactory results in most cases. In future studies, more attentions should be paid to exploring methodologies that tightly couple the carbon and water processes.

Moreover, some studies have shown that GPP and LUE increase with increasing diffuse radiation [94]. The diffuse radiation may reach shadowed leaves where the photosynthesis rate is usually limited [95]. A previous study found that the enhancement in LUE by diffuse radiation is significantly different in different ecosystems. However, crops showed less sensitivity to diffuse radiation than natural vegetation, which may be due to cultivation management by farmers [96].

We used GPP derived from flux tower sites to calibrate model parameters and verify the model results. GPP is not directly observed by the eddy covariance system. The partitioning algorithm for calculating GPP from $CO_2$ flux measurements introduces uncertainty. There are too few cropland flux tower sites globally, and these sites are not evenly distributed in space. Most of the sites are located in temperate and boreal regions (such as Europe and North America), and the semi-arid and tropical regions are underrepresented in the calibration dataset. These also bring large uncertainty to large scale applications. Furthermore, the pixel size (1 km in this study) of remote sensing data is inconsistent with the footprint of ground GPP observations (generally several hundred meters depending on the winds and atmospheric stability), which will inevitably lead to uncertainties in the verification of results. Further research needs to consider the impact of scale mismatch between ground measurements and the pixel size of satellite observations, and combine different spatial resolutions data for verifying model performance.

It should be noted that the satellite remote sensing-based FAPAR was used in our parameter optimization procedure, and uncertainty in the remote sensing FAPAR may be attributed to the uncertainty of the optimized parameters in some sites, which in turn generates uncertainty in the estimated GPP. Improvement on remote sensing FAPAR is necessary for increasing the accuracy in global cropland GPP estimation using LUE type models.

## 6. Conclusions

In this study, a light-use-efficiency-based model (EF-LUE) was developed and applied to estimate global cropland GPP from 2001 to 2019. The water availability variable (i.e., indicating moisture stress), expressed by evaporative fraction (EF), was integrated in the EF-LUE model to consider the impact of water availability on GPP. Three key model parameters ($\varepsilon_{max}$, $T_{opt}$, and $VPD_0$) in the developed EF-LUE model were optimized based on ground flux observations, and the optimized parameters were further extrapolated spatially according to climate zone classifications for global GPP estimation. The proposed EF-LUE model performed well in simulating spatial and temporal variations of global cropland GPP. It showed overall the highest $R^2$, KGE, and NSE and the lowest RMSE over the four existing GPP datasets (i.e., MOD17 GPP, revised EC-LUE GPP, GOSIF GPP, and PML-V2 GPP), when compared with the ground measurements of GPP from flux towers across crop sites globally. The accuracy of the EF-LUE GPP was also close to or even better than that of the PML-V2 GPP and GOSIF GPP, which highlighted the ability and potential of the LUE-based model for global GPP estimation, provided that the LUE model is well structured and calibrated, as with the EF-LUE model developed in this study. This study demonstrated the reliability of the EF-LUE model for estimating global cropland GPP by integrating EF as the indicator of soil water availability. It revealed that the EF-LUE GPP could track the impact of drought and heat-wave events on cropland productivity. This study provided a valuable dataset of cropland GPP at the global scale for the past two decades and could be further used for crop yield estimation.

**Author Contributions:** Conceptualization, D.D. and L.J.; methodology, D.D., L.J., Q.C. and C.Z.; software, D.D., C.Z. and M.J.; validation, D.D.; formal analysis, D.D., L.J., Q.C., C.Z., G.H., M.J. and J.L.; investigation, D.D., L.J., C.Z. and Q.C.; resources, L.J.; data curation, D.D., L.J. and C.Z.; writing—original draft preparation, D.D.; writing—review and editing, L.J. and C.Z.; visualization, D.D.; supervision, L.J., C.Z. and Q.C.; project administration, L.J.; funding acquisition, L.J. All authors have read and agreed to the published version of the manuscript.

**Funding:** This research was funded by the Strategic Priority Research Program of the Chinese Academy of Sciences (Grant No. XDA19030203), the National Key Research and Development Program of China (Grant No. 2017YFE0119100), and the National Natural Science Foundation of China (Grant Nos. 42001092, 42171039).

**Acknowledgments:** Authors thank data providers and platforms for GPP products from MOD17, the revised EC-LUE method, GOSIF method, NIRv method, and PML-V2; FAPAR data from the

Copernicus Global Land Service System (CGLS) GEOV2 dataset; ERA5 reanalysis data, ESA CCI landcover dataset; Köppen-Geiger climate classification data; and eddy covariance flux data from the communities including FLUXNET, ChinaFlux, AmeriFlux, AsiaFlux, and European Fluxes Database. We are grateful to everyone who contributed to these data. We thank the anonymous reviewers for their constructive comments.

**Conflicts of Interest:** The authors declare no conflict of interest.

## Appendix A

**Table A1.** The description of the Köppen-Geiger classes [56].

| Climate Type | Description |
|:---:|:---:|
| Af | Tropical, rainforest |
| Am | Tropical, monsoon |
| Aw | Tropical, savannah |
| BWh | Arid, desert, hot |
| BWk | Arid, desert, cold |
| BSh | Arid, steppe, hot |
| BSk | Arid, steppe, cold |
| Csa | Temperate, dry summer, hot summer |
| Csb | Temperate, dry summer, warm summer |
| Csc | Temperate, dry summer, cold summer |
| Cwa | Temperate, dry winter, hot summer |
| Cwb | Temperate, dry winter, warm summer |
| Cwc | Temperate, dry winter, cold summer |
| Cfa | Temperate, no dry season, hot summer |
| Cfb | Temperate, no dry season, warm summer |
| Cfc | Temperate, no dry season, cold summer |
| Dsa | Cold, dry summer, hot summer |
| Dsb | Cold, dry summer, warm summer |
| Dsc | Cold, dry summer, cold summer |
| Dsd | Cold, dry summer, very cold winter |
| Dwa | Cold, dry winter, hot summer |
| Dwb | Cold, dry winter, warm summer |
| Dwc | Cold, dry winter, cold summer |
| Dwd | Cold, dry winter, very cold winter |
| Dfa | Cold, no dry season, hot summer |
| Dfb | Cold, no dry season, warm summer |
| Dfc | Cold, no dry season, cold summer |
| Dfd | Cold, no dry season, very cold winter |
| ET | Polar, tundra |
| EF | Polar, frost |

## Appendix B

ETMonitor contains different modules to parameterize water flux components from soil-vegetation canopy, bare soil, open water, and ice and snow surfaces [32,33,64,65], and total ET (in a broader definition) is the sum of the components. The applications of ETMonitor can also be found in [65–67,91]. For this study, the results of ET in agroecosystems (soil-vegetation canopy) by ETMonitor were used. ETMonitor used the Shuttleworth-Wallace dual-source model [97], combined with parameterizations of a series of resistances, to estimate soil evaporation and vegetation transpiration for soil-vegetation canopy, and the total ET was calculated as the sum of soil evaporation and transpiration of crop plants. The soil evaporation ($E_s$) and plants transpiration ($T_v$) were estimated as

$$E_s = \frac{\Delta(Rn_s - G) + \rho c_p D_0 / r_a^s}{\lambda \Delta + \lambda \gamma (1 + r_s^s / r_a^s)} \tag{A1}$$

$$T_v = \frac{\Delta Rn_c + \rho c_p D_0 / r_a^c}{\lambda \Delta + \lambda \gamma (1 + r_s^c / r_a^c)} \tag{A2}$$

where $Rn_c$ (W·m$^{-2}$) is net radiation flux absorbed by the vegetation canopy; $Rn_s$ (W·m$^{-2}$) is net radiation flux arrives at the soil surface; $G$ is the soil heat flux (W m$^{-2}$); $r_a^c$ is the bulk boundary layer resistance of the vegetation (s m$^{-1}$), estimated according to the canopy height; $r_s^c$ is the bulk stomatal resistance of the canopy (s m$^{-1}$), estimated by Jarvis-type model [92,98]; $r_a^s$ is the aerodynamic resistance between the soil surface and the canopy

source height (s m$^{-1}$); $r_s^s$ is the surface resistance of the soil (s m$^{-1}$); $\Delta$ is the slope of the saturation vapor pressure curve of the air temperature (kPa K$^{-1}$); $\rho$ is the air density (kg m$^{-3}$); $c_p$ is the specific heat of air (J kg$^{-1}$ K$^{-1}$); $D_0$ is the water vapor pressure deficit at the canopy source height (kPa); $\lambda$ is the latent heat of evaporation (J kg$^{-1}$); $\gamma$ is the psychrometric constant (kPa K$^{-1}$). Details on ETMonitor and its regional applications can be found in our previous studies [32,33,64,65].

The net radiation (*Rn*) is estimated from the incoming and outgoing radiation fluxes in shortwave and longwave, and total *Rn* was partitioned to $Rn_c$ and $Rn_s$ following:

$$Rn = (1 - \alpha)R_{S\downarrow} + R_{L\downarrow} - R_{L\uparrow} - (1 - \varepsilon_s)R_{L\downarrow} \tag{A3}$$

where $\alpha$ is surface albedo; $R_{S\downarrow}$ is incoming shortwave radiation (W m$^{-2}$); $R_{L\downarrow}$ is incoming longwave radiation (W m$^{-2}$); $R_{L\uparrow}$ is outgoing longwave radiation (W m$^{-2}$); $\varepsilon_s$ is broadband surface emissivity. In addition, the interception loss amount was estimated by a revised Gash analytical model, and the details can be found in our previous study [99].

The daily *G* in ETMonitor was estimated by a machine learning algorithm established based on in situ measurements of *G* and several explanatory variables, and the results of *G* showed good agreement with the ground measurements at flux tower sites.

The ETMonitor was applied in combination with multi-source remote sensing data and re-analysis data from ERA5 to obtain global ET product at daily temporal and 1 km spatial resolutions in the last two decades. Applications of ETMonitor ET products can be found in [65–67].

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
