# Peer review of "Estimation of Global Cropland Gross Primary Production from Satellite Observations by Integrating Water Availability Variable in Light-Use-Efficiency Model"

_remotesensing, doi:10.3390/rs14071722_

Round 1

Reviewer 1 Report

  1. The abstract needs four elements. The abstract is also needed to revised thoroughly. The main quantitative results need to be shown.
  2. Line 123, the unit of the PAR is wrong, the time unit is needed.
  3. Line 126, How to get the value of 0.48?
  4. Table 2, Mazie would be maize.
  5. Figure 1 is not standardized.
  6. Table 4, the site is inaccurate. It is the climate type.
  7. Table 4, εmax is hard to 3.99.
  8. Figure 2, Simulated would be estimated.
  9. Figure 2, Figure 4, The estimated GPP values are obviously underestimated.
  10. Table 5 and Figure6, The authors need compare the EF-LUE with GOSIF and PML.
  11. Discussion need focus on the EF-LUE model.

Reviewer 2 Report

This MS parameterized a LUE model to estimate GPP of global crop. With so many GPP model, especially the model based on SIF, NIR, the GPP could be well estimated by present models. Therefore, this MS is not so interesting as my view.

EF was applied to quantify available soil moisture, which was demostrated by previous studies, such as Nutini et al. (2014). However, the essence of this method is to estimate evapotranspiration through remote sensing. Though the study on evapotranspiration has been published by the authors, the paper on IEEE was very difficulty to be accessed and the ET method could not be well qualified presently. Therefore, as my suggestion, the ET method should be clearly introduced in this MS, maybe as an appendix.

Reviewer 3 Report

The manuscript provided the development and evaluation of a light use efficiency (LUE) model to estimate   gross primary productivity (GPP). The developed model was based on optimizing parameters using data from ground-based eddy covariance CO2 and introducing the use of evaporative fraction. The resulting model was then generalized/applied at a global scale. The topic is interesting. However, the manuscript has major deficiencies related to methods, experiment design, and style. Some of these issues are provided below and should serve as guidance to improve the manuscript. The purpose of/objectives of the study were clearly listed but were not addressed properly.

Line 41-50: The three categories of GPP models: the process-based, empirical-based are district categories. However, referring to LUE-based models as a separate category is no rationale. The LUE could be considered process-based, empirical, or a combination. LUE is a methodology (or approach) but not a distinct category. Please reconsider how the third category was identified in this manuscript.  

Line 55: Please use “using” instead of from. The LUE model can use data from different sources. In this case EC-LUE is not a different type of LUE model but an indication it uses EC data.

Line 65-67: This a good reasoning of MODIS GPP. This interoperation could also be applied to the developed EF-LUE as Figure 4. shows similar behavior. Please refer to related commented later on in this review.

Line 109-114: The purpose of/objectives of the study were clearly listed but were not addressed properly. Optimizing the parameters and assessing the performance of the EF-LUE could be separate objective … which wasn’t addressed in a way to show how significantly different the optimized parameters from what’s currently sed.

Line 161: provide a description of how the Fsm without EF is calculated.

Line 198 – 205: please provide all website links as citations and place their references in the “Refences” section.

Line 226: Please provide a brief description of the ETMonitor. Does it provide LE, H, G, and Rn. Providing a reference for the mode is not enough.

Figure 1: Please put a number next to each star and provide a legend with the name of each site.

Table 4:  This is the first time in the manuscript these sites were mentioned. There is no information about their location, why they were selected, data period. What’s the rationale that optimized parameters based on these sites can be generalized globally. Actually, based on Table 4, it seems that there are no significant differences between the parameters of (CRO/Dfa) the LUE models with and without the use of EF. Which formula/equation/model was used to estimate Fsm in the LUE model without EF.

Line 280: How the seasonal variation was identified (also see Line 293). These percentages are based on statistical metrics.

Line 302: What is common in these sites?

Figure 4: Place legend on the figure. Visually estimates of GPP with and without EF are not significantly different.

Table 4: What do you mean by “Revised” EC-LUE. Did the authors update this model as well? Also, R2 is not a good statistical indicator to use to evaluate/compare models’ performance. It can provide high values while RMSE could also be high. For example, check (CH-Oe2 and FR-Gri). This could also be the case with KGE. Try to use other statistical metrics such as Nash & Sutcliffe efficiency coefficient.

Line 354: How this pattern “shows a similar spatial pattern” was evaluated (visually or using statistical indicator)

Line 360: where this overestimation was observed. Based on Table 5 (BIAS) it is mostly underestimation.

Figure 5: The maps are good in showing the spatial variability, but they lack showing where and how the different estimates of GPP based on these models are. Maybe providing maps with differences in GPP is more appropriate.

Line 375: “selected” This selection based on what criteria?

Line 377 – 381: these should have been mentioned in the methods sections

Figure 6: What's the purpose of this comparison? if models provide higher or lower GPP than other models that doesn’t give indication which model is more accurate than the others.

Reviewer 4 Report

Dear Authors,

the modeling of GPP is quite interesting for many ecological applications. Anyway is completely missed the retrieval of these indicator with the help of other sensors and approaches at global scale .I suggest to read and integrate your speech, citing it, in the introduction with other interesting papers regarding the modeling of GPP and LUE using not only data coming from the tower but also microwave remote sensing information from other sensors, enlarging the field of retrieval of these two important ecological indicators.  One that comes into my mind is :

Vaglio Laurin, Gaia, et al. "Monitoring tropical forests under a functional perspective with satellite‐based vegetation optical depth." Global Change Biology 26.6 (2020): 3402-3416.

Respect to the other sections, in Results I suggest to split Fig. 2 in summer period and winter, separating the statistics to these periods and if possible in climatic zones too.

Fig. 4 I'd like to see enlarged trends in order to see if there is a lag between the two models and the eventual explanation about.

I suggest major revision.

Also, the following studies are about the estimation of  gross primary production and carbon sink by using microwave radiometry:

“Daily estimation of gross primary production under all sky using a light use efficiency model coupled with satellite passive microwave measurements”, by Wang et al., 2021

“A carbon sink-driven approach to estimate gross primary production from microwave satellite observations”, by Teubner et al., 2019

“Mean European carbon sink over 2010–2015 estimated by simultaneous assimilation of atmospheric CO2, soil moisture, and vegetation optical depth”, by Scholze et al., 2019

“Assessing the relationship between microwave vegetation optical depth and gross primary production”, by Teubner et al., 2018

Round 2

Reviewer 1 Report

The Authors had revised the manuscript according to my comments.

Author Response

Thank you for reviewing the manuscript.

Reviewer 2 Report

On the suggestion to describ ET was adopted but almost no information was given. Becuase those two equations involved lots of parameters that were not explained how to quantify them. Given very complex process on ra and rs, the methods would be invovled a very complex ecological process, which would bring lots of uncertainties into the estimation of GPP. As a suggestion, the relationship between the ra, rs with GPP should be discussed, that is, as interaction between water and carbon cycle in terrestrial ecosystems.
